# Exponentially-enhanced quantum sensing with many-body phase transitions

Saubhik Sarkar [1,2], Abolfazl Bayat [1,2,3] ✉, Sougato Bose[4] ✉ & Roopayan Ghosh [4] ✉

Quantum sensors based on critical many-body systems are known to exhibit enhanced sensing capability. Such enhancements typically scale algebraically with the probe size. Going beyond algebraic advantage and reaching exponential scaling has remained elusive when all the resources, such as the preparation time, are taken into account. In this work, we show that many-body systems featuring first order quantum phase transitions can indeed achieve exponential scaling of sensitivity, thanks to their exponential energy gap closing. Remarkably, even after considering the preparation time using local adiabatic driving, the exponential scaling is sustained. Our results are demonstrated through comprehensive analysis of three paradigmatic models exhibiting first order phase transitions, namely Grover, $p$-spin, and biclique models. We show that this scaling survives moderate decoherence during state preparation and also can be optimally measured in experimentally available basis. Our findings comply with the fundamental bounds and we show that one can harness the exponential advantage through an adaptive strategy even away from the phase transition point.

Quantum sensing is an important component of quantum technologies due to its potential for developing a new generation of probes, capable of environmental monitoring with unprecedented precision beyond classical sensors[1]. In this context, the sensitivity of a probe can be quantified by Fisher information, inverse of which puts a bound on the uncertainty of the estimation protocol[2,3]. In classical sensors, Fisher information, at best, scales linearly with resources, such as the system size $L$ (standard limit). Quantum features may result in super-linear scaling of Fisher information, known as quantum enhanced sensitivity. This has been discovered in a series of seminal works by Giovannetti et al., where they showed that a special form of entangled states, known as the Greenberger-Horne-Zeilinger (GHZ) states, can be used to estimate the phase imprinted by a unitary operation with Fisher information scaling as $L^2$ (Heisenberg limit)[4]. In the presence of $k$-body interactions in the generator of the unitary operation, the sensitivity can be further enhanced to[5] $L^{2k}$. In a

fundamentally different approach, quantum enhanced sensitivity has also been identified in many-body systems[6] when they go through a quantum phase transition. This includes, first-order[7–9], second-order[10–16], Floquet[17], time crystal[18–20], Stark[21] and quasi-periodic[22] localization, and topological[23,24] phase transitions. In all these critical systems, where Fisher information scales algebraically as $L^\beta$ (with $\beta > 1$), the many-body system goes through an algebraic energy gap closing in its spectrum. This gives rise to the conjecture that energy gap closing might be the reason behind quantum enhanced sensitivity[25], which is supported by a recent seminal work[26] on metrological limits. Non-equilibrium quench dynamics in many-body systems have also been explored for achieving quantum-enhanced sensitivity[27] in which Fisher information also depends on evolution-time $t$ and typically scales as $t^2 L^\beta$, following the scope of the generalized Heisenberg limit[5,28]. While in all these cases, Fisher information, and thus the precision, scales algebraically, one may wonder

[1]Institute of Fundamental and Frontier Sciences, University of Electronic Science and Technology of China, Chengdu 611731, China. [2]Key Laboratory of Quantum Physics and Photonic Quantum Information, Ministry of Education, University of Electronic Science and Technology of China, Chengdu 611731, China. [3]Shimmer Center, Tianfu Jiangxi Laboratory, Chengdu 641419, China. [4]Department of Physics and Astronomy, University College London, Gower Street, WC1E 6BT London, UK. ✉e-mail: abolfazl.bayat@uestc.edu.cn; s.bose@ucl.ac.uk; ucaprgh@ucl.ac.uk

whether quantum features can result in the possibility of even a better quantum advantage, namely exponentially enhanced quantum sensing.

Exponential enhancement has in fact been reported in ref. 29, for the GHZ-based sensing protocols where the required entanglement in the initial state demands exponentially large number of unitary gates, making its implementation very challenging. In non-Hermitian systems exponential sensitivity can be achieved in the eigenenergy spectrum at exceptional points (parameter value where multiple eigenvalues and eigenstates coalesce)[30–34]. However, it is debated whether the quantum advantage would survive the quantum noise arising from the non-orthogonality of the eigenstates[35,36]. Proposals based on tight-binding non-Hermitian topological systems have also reported exponential sensitivity[37–39] for inferring the value of a perturbative boundary coupling in the steady state. While these works show great potential for quantum enhancement, the schemes are restrictive for several reasons: (i) the preparation time for the steady state is typically long whose consideration in resource analysis may destroy quantum advantage; (ii) the schemes are limited to driven coupled resonators as non-Hermitian Hamiltonians cannot faithfully describe an open system evolution beyond a short time; and (iii) the necessity for measuring a perturbatively small coupling exclusively at the boundary is also a big constraint. In fact, a fundamental constraint derived in ref. 40 show that non-Hermitian sensors cannot perform better than Hermitian counterparts. Therefore, finding a concrete protocol with Hermitian systems showing exponential scaling advantage even when the resources are taken into account is highly desirable.

In this work, we show that it is indeed possible to achieve the exponential scaling for sensitivity by leveraging the first-order phase transitions where the energy gap also closes exponentially in system size. We then show that even if the preparation time of the critical state is taken into account, the exponential sensitivity still prevails. This can be intuitively understood from the aforementioned bound[5,28] bearing the quadratic scaling in time which itself grows exponentially with system size. Our results are shown analytically for a paradigmatic model, namely Grover model, and numerically for $p$-spin and a biclique spin model that are prototypical systems from a quantum annealing perspective. The results satisfy the fundamental bounds of quantum sensing schemes, and the estimation process can be performed in experimentally available measurement basis. We consider the issue of decoherence during state preparation and show that the exponential scaling is sustained up to certain dephasing strength. The local nature of criticality-based sensors is also addressed and an adaptive estimation strategy is sketched out to harness the full advantage of the exponential scaling for arbitrary value of the parameter to be estimated.

## Results

### Parameter estimation

In this work, we will be considering single parameter estimation, where the value of an unknown parameter $\theta$ is estimated by performing measurements on a quantum state $\rho(\theta)$ that encodes the parameter. The quantum state is known as the probe state and the measurement outcomes are fed into an estimator function to infer the value of the parameter. In general, the measurement can be described by a complete set of Positive Operator Valued Measurement (POVM) $\{\Pi_n\}$ where the $n$th outcome occurs with probability $p_n(\theta) = \text{Tr}\left[\rho(\theta)\Pi_n\right]$. The uncertainty of estimating the unknown parameter $\theta$, quantified by standard deviation $\delta\theta$, is bounded through Cramér-Rao inequality $\delta\theta \geq 1/\sqrt{MF^C}$. Here, $M$ is the total number of measurements and the basis-dependent classical Fisher information (CFI) is[3] $F^C = \sum_n p_n(\partial_\theta \log p_n)^2$. In order to have a measurement-independent quantity, one can maximize the CFI with respect to all possible measurements to obtain Quantum Fisher Information (QFI) $F^Q$, namely $F^Q = \max_{\{\Pi_n\}} F^C$. As a result, the Cramér-

Rao inequality becomes

$$\delta\theta \geq \frac{1}{\sqrt{MF^C}} \geq \frac{1}{\sqrt{MF^Q}}, \tag{1}$$

where QFI gives the ultimate precision limit of the estimation. Interestingly, for evaluating the QFI one can avoid the notorious optimization over all possible measurement basis and instead consider the symmetric logarithmic derivative (SLD) operator $\mathcal{L}$, implicitly defined as

$$\frac{\partial\rho(\theta)}{\partial\theta} = \frac{\rho(\theta)\mathcal{L}_\theta + \mathcal{L}_\theta\rho(\theta)}{2} \tag{2}$$

The QFI is then expressed as $F^Q = \text{Tr}\left[\rho(\theta)\mathcal{L}_\theta^2\right]$. For pure states $\rho(\theta) = |\psi(\theta)\rangle\langle\psi(\theta)|$, the expressions are simplified to $\mathcal{L}_\theta = 2\partial_\theta\rho(\theta)$, and consequently[3]

$$F^Q(\theta) = 4\left(\langle\partial_\theta\psi(\theta)|\partial_\theta\psi(\theta)\rangle - |\langle\partial_\theta\psi(\theta)|\psi(\theta)\rangle|^2\right). \tag{3}$$

As QFI quantifies the rate of change of the probe state, it is also equivalent to the fidelity susceptibility. In the context of the ground state of a Hamiltonian $H(\theta)$, this leads to another expression for QFI[41]

$$F^Q(\theta) = 4\sum_{n\neq0} \frac{|\langle\psi_n(\theta)|\partial_\theta H(\theta)|\psi_0(\theta)\rangle|^2}{(E_n(\theta) - E_0(\theta))^2}. \tag{4}$$

Here $|\psi_n\rangle$ and $E_n$ are the $n$-th eigenvector and eigenvalue of $H(\theta)$. It is worth emphasizing that to achieve the ultimate precision limit, given by the QFI, one has to perform measurement in the optimal basis. The optimal measurement basis is not unique, although one choice is always given by the projectors formed from the eigenvectors of the SLD operator $\mathcal{L}_\theta$.

### Fundamental QFI bounds in many-body probes

While in the Cramér-Rao inequality (Eq. (1)) the estimation precision, quantified by standard deviation, is bounded by $1/\sqrt{F^Q}$, there has been an interest to find analytical bounds on the QFI. Such bounds are quite insightful to give us a hint for the best possible scaling of the QFI. These bounds have been established for various scenarios, including non-equilibrium dynamics[5], ground state of many-body Hamiltonians[26], and steady state sensing[28]. In particular, we are concerned with the ground state probe, for which the upper bound of QFI has been derived recently[26] to concretely prove the connection with both the energy gap and the spectral properties of the Hamiltonian. For Hamiltonians in the form $H(\theta) = H_C + H_\theta$, with a control term $H_C$ and the parameter dependent term $H_\theta$, the upper bound of QFI of the ground state is given by[26],

$$F^Q(\theta) \leq \frac{||\partial_\theta H_\theta||^2}{\Delta^2}, \tag{5}$$

where the operator seminorm is the difference between the maximum and minimum eigenvalues, $||\partial_\theta H_\theta|| = \lambda_{\max} - \lambda_{\min}$ and $\Delta$ is the energy gap between the ground state and the first excited state. Both these terms typically display scaling behaviour in system size near critical points, which controls the scaling of the upper bound. Thus, the ultimate scaling of the QFI may be determined by the individual scaling behaviour of these two terms.

On the other hand, when the probe state is prepared dynamically by evolving a suitably chosen initial state with a Hamiltonian consisting of a time-dependent control term in the form $H(\theta, t) = H_C(t) + H_\theta$, the upper bound of QFI is given by the generalized Heisenberg limit[5,28]

$$F^Q(\theta, t) \leq t^2 ||\partial_\theta H_\theta||^2. \tag{6}$$

Note that Eq. (6) is valid for any dynamical scenario, including the adiabatic state preparation. In such schemes, the time needed for adiabatic preparation of the probe in the ground state of a complex Hamiltonian can be made inversely proportional to the minimum energy gap, namely ~ $1/\Delta$[42]. Hence, by replacing time $t$ with $1/\Delta$ in Eq. (6) one can effectively see the connection with the bound given in Eq. (5).

It is worth emphasising that both the Eqs. (5) and (6) only impose an upper bound on the QFI. In fact, while these bounds are very insightful for capturing the scaling in many-body probes, they usually overestimate the value of the QFI, which is the relevant quantity for determining the achievable precision. Indeed, for the particular systems considered in this work, the QFI near criticality expectedly follows the bound but does not saturate it in general.

## Models

Quantum many-body systems have been proven to be very useful to serve as quantum sensors achieving quantum enhanced sensitivity in both equilibrium and non-equilibrium scenarios[25]. In particular, the ground state of many-body systems across various types of phase transitions have been identified as effective quantum sensors. In such systems the Hamiltonian, in general, has the form

$$H(\theta) = H_1 + \theta H_2, \tag{7}$$

where $H_1$ and $H_2$ are two competing terms and $\theta$ is the unknown parameter to be estimated. The $H_2$ component therefore serves as the derivative terms in Eqs. (4), (5), and (6). When the role of competing terms become comparable, say at $\theta = \theta_c$, the system may go through a phase transition where the ground state $|GS(\theta)\rangle$ changes dramatically. From the spectral perspective, the ground state and the first excited state go through an anti-crossing at $\theta = \theta_c$ where the energy gap vanishes in the thermodynamic limit. If the energy gap closes exponentially with the system size, then the system goes through a first order phase transition in which the order parameter discontinuously jumps across the transition point. On the other hand, if the energy gap closes algebraically, then the order parameter changes continuously and it is the first derivative that becomes non-analytic at the phase transition. While the capability of utilizing second order phase transitions as effective quantum sensors has been fully characterized[12], the first order phase transitions have not been completely explored. As we shall see in the following sections, first order phase transitions indeed allow for estimating $\theta$ with exponential sensitivity, quantified by exponential scaling of QFI with the system size. In the following we introduce three paradigmatic models with first order phase transitions, namely Grover, p-spin, and biclique spin systems.

## Grover model

We first consider a system consisting of $L$ qubits which span a Hilbert space of dimension $N = 2^L$. Every qubit configuration can coherently tunnel to another with equal probability, though one specific qubit configuration $|m\rangle$ has a different energy from the rest. In this situation one can write the Hamiltonian,

$$H_{\text{Grover}}(\theta) = -|m\rangle\langle m| - \theta|\psi\rangle\langle\psi|, \tag{8}$$

where

$$|\psi\rangle = \frac{1}{\sqrt{N}}\sum_{j=1}^{N}|j\rangle = \frac{1}{\sqrt{N}}|m\rangle + \sqrt{\frac{N-1}{N}}|m^\perp\rangle, \tag{9}$$

with

$$|m^\perp\rangle = \frac{1}{\sqrt{N-1}}\sum_{j\neq m}|j\rangle. \tag{10}$$

One can easily show that the Hamiltonian in Eq. (8) can be effectively be written as a two level system spanned by $|m\rangle$ and $|m^\perp\rangle$ as,

$$H_{\text{Grover}}(\theta) = -\begin{pmatrix} \frac{\theta}{N}+1 & \frac{\theta\sqrt{N-1}}{N} \\ \frac{\theta\sqrt{N-1}}{N} & \frac{\theta(N-1)}{N} \end{pmatrix}. \tag{11}$$

This model is analytically tractable and will serve as a robust theoretical foundation for our conclusions. In this representation, the first-order phase can be analytically shown to be occurring at $\theta_c = 1$[42].

## p-spin model

The second model we consider is based on p-spin model[43,44], in a system of $L$ qubits, represented by,

$$H_{p-\text{spin}}(\theta) = \left[ -\lambda L^{1-p}\left(\sum_{j=1}^{L}\sigma_j^z\right)^p + (1-\lambda)L^{1-k}\left(\sum_{j=1}^{L}\sigma_j^x\right)^k \right] - \theta\sum_{j=1}^{L}\sigma_j^x \tag{12}$$

where, $p$ and $k$ are integer numbers and $0 \le \lambda \le 1$ is an external parameter that tune the system to feature either first or second order phase transition. For $\lambda = 1$, one gets back the traditional $p-$spin model, in which one has a first order phase transition for $p \ge 3$. By choosing increasing values of $p \ge 3$ for $\lambda = 1$, it is possible to shift the critical point from $\theta_c = 1.3$ for $p = 3$ towards $\theta_c = 1$ for $p \to \infty$ which corresponds to the Grover model[45]. For $\lambda \neq 1$, we have an additional antiferromagnetic fluctuation term[46], i.e. the middle term in Eq. (12), which can change the first order phase transition to a second order one. For instance by choosing $\lambda = 0.1$, $p = 5$, $k = 2$ one observes a second order quantum phase transition at $\theta_c = 1.8$[47]. Due to degeneracy issues with even $p$, we shall only consider the odd cases in this work.

## Biclique spin system

Finally, we consider a biclique graph that can be easily implemented on existing quantum hardware and has been utilized in studies of maximum weighted independent set (MWIS) problems[48,49]. In such graphs, the system is partitioned into two subsystems $A$ and $B$ with $L_A$ and $L_B$ spins, respectively. We consider $L_A = L_B + 1$ which means the total system size will be $L = 2L_A + 1$. Every spin in the subsystem $A$ interacts with every spin in subsystem $B$ with antiferromagnetic Ising interaction with strength $J$. In addition, the two subsystems are affected by two different uniform magnetic fields $h_A$ and $h_B$. To induce a competing term the whole system is subjected to a uniform transverse magnetic field. The Hamiltonian can be expressed as[49]

$$H_{\text{Biclique}} = \left[ J\sum_{j_A=1}^{L_A}\sum_{j_B=1}^{L_B}\sigma_{j_A}^z\sigma_{j_B}^z + h_A\sum_{j_A=1}^{L_A}\sigma_{j_A}^z + h_B\sum_{j_B=1}^{L_B}\sigma_{j_B}^z \right] + \theta\sum_{j=1}^{L}\sigma_j^x. \tag{13}$$

By tuning the longitudinal magnetic fields $h_A$ and $h_B$ one can engineer the emergence of a first order phase transition at different values of $\theta_c$.

## Scaling analysis

Now we discuss the sensing capabilities of the three models introduced in the previous section to estimate $\theta$ in the ground state due to phase transition. We focus on the scaling of two quantities with respect to the system size. First, we consider the scaling of the energy gap which is necessary to characterize the type of the phase transition. Second, we analyze the scaling of the QFI as a figure of merit for the sensing capability of our models.

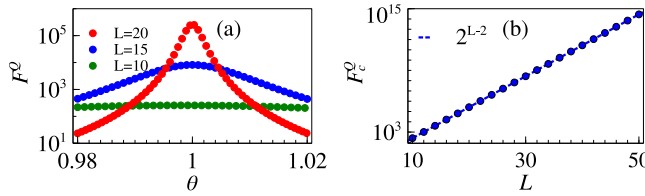

**Fig. 1 | Sensing with Grover model. a** QFI around the critical point for different system sizes. **b** QFI scaling at criticality ($\theta_c = 1$). The dotted line shows the asymptotic QFI value.

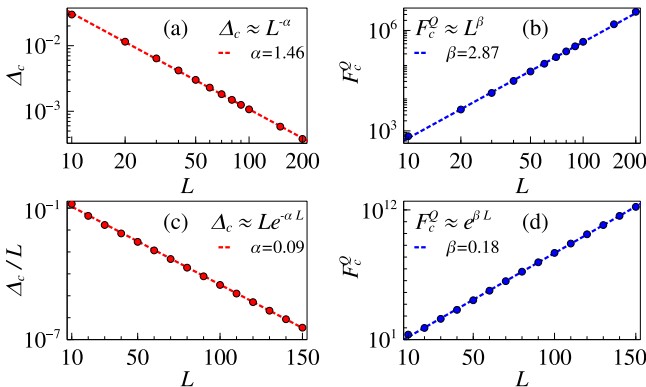

**Fig. 2 | Sensing with $p$-spin model. a** Energy gap scaling for $p$-spin model (Eq. (12)) for $\lambda = 0.1$, $p = 5$, $k = 2$ at criticality occurring near $\theta = 1.8$. **b** Algebraic QFI scaling at criticality. **c** Energy gap scaling for $\lambda = 1$, $p = 3$ at criticality occurring near $\theta = 1.3$. **d** Exponential QFI scaling at criticality for this case.

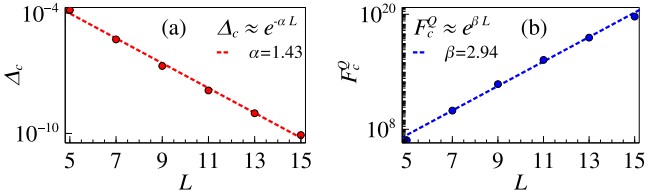

**Fig. 3 | Sensing with biclique model. a** Energy gap scaling for the biclique spin system (Eq. (13)) at criticality occurring near $\theta_c = 0.05$ with $h_{A[B]} = \left(L_{B[A]}J - 2\frac{W_{A[B]}}{L_{A[B]}}\right)$ with $J = 1$, $W_A = 0.49J$ and $W_B = 0.5J$. **b** QFI scaling at criticality for this system.

## Sensing with Grover model

For the Grover model, one can obtain the eigenspectrum analytically to compute the energy gap,

$$\Delta(\theta, N) = \frac{\sqrt{N^2(1-\theta)^2 + 4N\theta}}{N}. \tag{14}$$

Note that $N = 2^L$ is the Hilbert space size. The energy gap $\Delta$ has a minimum at $\theta = \theta_c = 1$ with $\Delta_c = \Delta(\theta_c, N) = 2/\sqrt{N} = 2^{1-\frac{L}{2}}$. For the ground state of the system one can compute the QFI with respect to $\theta$ which takes the form

$$F^Q(\theta) = \frac{4(N-1)}{[N(1-\theta)^2 + 4\theta]^2}. \tag{15}$$

The peak structure of QFI around the critical point is shown in Fig. 1a. As the system approaches its critical point, the QFI becomes $F_c^Q = F^Q(\theta_c) = (N-1)/4 \approx 2^{L-2}$ in large $L$ limit. This exponential scaling of $F_c^Q$ is numerically verified in Fig. 1b which shows that the asymptotic behavior is captured by finite number of qubits as well. We also

observe the critical exponent for the QFI growth is twice of that for the gap decrease.

It is also informative to verify the bound on the QFI given by Eq. (5). Interestingly, the QFI bound is almost saturated at the critical point for large system sizes. Here, $\|H_2\| = 1$, and the QFI can be shown analytically to obey the bound (see the Methods section). In this model the scaling of both the QFI and the bound is merely determined by the scaling of the energy gap.

## Sensing with $p$-spin model

The second model that we consider for sensing is the $p-$ spin model, introduced in Eq. (12). In this model, not only the critical point $\theta_c$ can be tuned by controlling $p$, $k$, and $\lambda$, but also the nature of phase transition can be controlled. For example, for $p = 5$, $k = 2$ and $\lambda = 0.1$, the phase transition is of second order type and happens at[47] $\theta_c = 1.8$. To show this, in Fig. 2a, we plot the energy gap as function of system size at criticality. As the figure shows, the energy gap closes algebraically, i.e. $\Delta_c \propto L^\alpha$ with $\alpha \approx 1.46$, signaling the second order nature of the phase transition. The corresponding QFI at the critical point is also plotted as a function of system size $L$ in Fig. 2b. Clearly, the QFI shows an algebraic scaling i.e. $F_c^Q \propto L^\beta$ with $\beta \approx 2.87$ which is the conventional behavior at the second order quantum phase transitions. Note that we again observe that $\beta \sim 2\alpha$.

By tuning $\lambda = 1$ and $p = 3$ one can observe a first order phase transition at[45] $\theta_c = 1.3$. The energy gap in this case is known to close exponentially with a multiplicative correction term, so that[45] $\Delta_c \sim Le^{-\alpha L}$. As shown by the numerical fit in Fig. 2c, $\alpha \sim 0.09$. The corresponding ground state QFI at the critical point exponentially grows with $L$, i.e. $F_c^Q \sim e^{\beta L}$ with $\beta \approx 0.18$, as shown in Fig. 2d.

The relation $\beta \sim 2\alpha$ can be explained by the equivalence between QFI and fidelity susceptibility in Eq. (4). At criticality, the dominant contribution in the sum on the right hand side of the Eq. (4) comes from the first term (with the first excited state) and the overlap in the numerator were found to be linearly scaling with system size. This cancels the linear multiplicative scaling factor of the gap in the denominator and consequently $\beta = 2\alpha$.

One can also verify this by considering the scaling of the bound in Eq. (5). In this model, one can show that $\|H_2\| = 2L$, which implies that this term also contributes to the scaling of the bound and cancels the linear scaling factor that appears in the energy gap as well at the critical point. Consequently, both the bound and the QFI scales purely exponentially with respect to the system size. Unlike the Grover model, the bound is not saturated near criticality in $p$-spin model, despite having the same scaling behaviour (see the Methods section). This arises due to different prefactors for the bound and the computed QFI.

## Sensing with biclique spin model

Now we focus on the sensing capacity of the biclique spin model described in Eq. (13). Following the recipe of refs. 48,49, we take $h_{A[B]} = \left(L_{B[A]}J - 2\frac{W_{A[B]}}{L_{A[B]}}\right)$, with $J = 1$, $W_A = 0.49J$ and $W_B = 0.5J$. For these choices of parameters, the first order quantum phase transition takes place at $\theta_c \approx 0.05$. In Fig. 3a we plot the scaling of energy gap at the critical point, namely $\Delta_c$, with respect to system size $L$. We observe an exponential falling off $\Delta_c \sim e^{-\alpha L}$ with exponent $\alpha \approx 1.43$. Consequently, the corresponding ground state QFI at the critical point exponentially grows with systems size as $F_c^Q \sim e^{\beta L}$ with exponent $\beta \approx 2.94$, as displayed in Fig. 3b. The observation of $\beta \sim 2\alpha$ applies here also.

In biclique spin model, the scaling analysis is limited to small system sizes as large number of spins cannot be handled by exact diagonalization method. Regarding the bound in Eq. (5), one can see that $\|H_2\| = 2L$. The scaling of the upper bound therefore consists of an extra linear factor, along with the exponential size dependence coming from the energy gap, which is obtained through finite-size numerics. Thus, the QFI is shown numerically to obey the bound predicted by Eq. (5) (see the Methods section), although the bound is never saturated.

## Resource analysis

So far, we have considered system size as the only resource for sensing. However, since we focus on the ground state QFI, we need to first prepare the ground state of the corresponding Hamiltonians. Typically, there are two ways to prepare a many-body system in its ground state: (i) cooling to ground state; and (ii) adiabatic state preparation. Since the energy gap closes exponentially, both of these methods face severe challenges as cooling will be affected by critical slowing down and adiabatic state preparation requires extremely long preparation times. One may also consider the preparation time as a resource for accomplishing the sensing task. In order to incorporate time into resource analysis, one may consider the total time $T_{tot}$ the is used for collecting the data through probe preparation and measurement. If the preparation of the probe takes time $T$, within the available total time one can get $M = T_{tot}/T$ number of measurement. By inserting this into Eq. (1) one gets $\delta\theta \geq 1/\sqrt{T_{tot} F^Q/T}$. This immediately suggests that for incorporating the total time as a resource, one has to consider the rescaled QFI, i.e. $F^Q/T$, as the new figure of merit. The rescaled QFI has long been used for resource analysis in various works[50–54].

While both cooling and adiabatic state preparation are affected by closing of the energy gap, for sake of simplicity we shall only focus on adiabatic state preparation in this work. The adiabatic theorem states that to prepare the ground state of a many-body system one can start with an easily preparable ground state of a simple Hamiltonian and slowly change the Hamiltonian into the desired one. If the evolution is slow enough, taking place over a long time $T$, then quantum state of the system follows the ground state of the instantaneous Hamiltonian and thus reach the desired ground state at the end of the evolution. The original formulation of the adiabatic theorem requires that $T \sim 1/\Delta_{min}^2$ where $\Delta_{min}$ is the minimum energy gap of the Hamiltonian throughout the evolution[55]. However, there has been a lot of effort to speed up the state preparation[42,56–58]. In fact, it has been demonstrated that one can reach the ground state with high fidelity even if the evolution time only scales as $T \sim 1/\Delta_{min}$[42].

In order to analyze preparation time in our schemes, we re-parameterize the Hamiltonian in Eq. (7) into the following time-dependent form

$$H(s(t)) = s(t)H_1 + (1 - s(t))H_2, \quad (16)$$

where the parameter $\theta$ is now equivalent to $(1 - s(t))/s(t)$. The parameter $s$ evolves from 0, where the probe is initialized in the ground state of $H_2$, to a value corresponding to the desired $\theta$. The minimum energy gap happens at $\theta = \theta_c$. Therefore, it is plausible to make the preparation time scale as $T \sim 1/\Delta_c$. As we have shown already, the QFI typically scales as $F_c^Q \sim e^{\beta L}$ and the energy gap closes as $\Delta_c \sim e^{-\alpha L}$. Consequently, our new figure of merit $F_c^Q/T \sim e^{(\beta - \alpha)L}$. Remarkably, as demonstrated in all examples, we universally observe $\beta \sim 2\alpha$ which results in $F_c^Q/T \sim e^{\beta L/2}$, signaling exponential advantage even when the preparation time is included in our resource analysis.

To verify the above statement, we numerically prepare the ground state of each of the three models described before using local adiabatic driving[42], which results in $T \sim 1/\Delta_c$. We start with $s = 0$, i.e. the ground state of $H_2$, and then evolve $s$ with time over a long time interval $T$ using a particular *schedule* $s(t)$. This choice of time-dependent $s(t)$ for local adiabatic driving needs to be fast when the system is far from criticality and slow near the critical point. To get the quantum state at each time one has to solve the Schrodinger equation

$$i\hbar \frac{\partial|\psi(t)\rangle}{\partial t} = H(s(t))|\psi(t)\rangle, \quad (17)$$

with the initial state $|\psi(0)\rangle$ being the ground state of $H_2$.

For the Grover model, it can be analytically shown that[42] $s(t) = N \left( \tan^{-1}\sqrt{N-1}(2s-1) + \tan^{-1}\sqrt{N-1} \right)/(2\epsilon\sqrt{N-1})$. This results in $T = \pi/2\epsilon\Delta_c$ where the fidelity between the state of the probe and the instantaneous ground state, namely $\mathcal{F}(t) = |\langle GS(t)|\psi(t)\rangle|^2$, is lower bounded as $\mathcal{F}(t) \geq (1 - \epsilon^2)$. We have numerically verified this in Fig. 4a, where we plot the fidelity $\mathcal{F}$ versus $\theta$ for a system of size $L = 20$. As the figure shows, one can achieve a fidelity of 0.99 at the critical point. Furthermore, we compare the variation of $F^Q$ across $\theta_c$ for the exact ground state $|GS(s(t))\rangle$ in Fig. 4b and the prepared state $|\psi(s(t))\rangle$ in Fig. 4c. We observe that for that the small loss of fidelity has very little effect on QFI, which indicates that the exponentially effective quantum sensing at the first order critical point in the Grover model survives under local adiabatic state preparation. For the other two models the schedule $s(t)$ was derived numerically using local adiabatic driving and the total preparation time was expectantly found to be bounded by $1/\Delta_c$ (see the Methods section). The corresponding results for the $p$-spin model are shown in Fig. 4d–f, where we observe results similar to the previous case. For the biclique model, as shown in Fig. 4g, the local adiabatic evolution results in the fidelity going below 0.98 near the critical point $\theta_c$ out of the three systems. Correspondingly, we observe that there is an increase in $F_Q$ for the ground state prepared by local adiabatic evolution compared to the exact ground state. It turns out that for small system sizes, the minuscule excitations above the true instantaneous ground state caused by the time evolution results favourably for the QFI.

Having established the fact that the critical QFI scales exponentially even after taking the adiabatic preparation time into account, we now give a concrete framework to create the probe state of for an unknown $\theta$, which is the realistic sensing scenario. Without loss of generality, we assume that the sensing apparatus is designed to detect a non-negative $\theta$, and consequently, its dynamics is governed by Eq. (7). As we know $H_1$ and $H_2$, we can determine the critical parameter $\theta_c$, while $\theta$ still remains unknown. We then apply a time-dependent control field $s(t)/(1 - s(t))$ to the $H_1$ component and a critical field $\theta_c$ to the $H_2$ component, so that the total Hamiltonian becomes

$$H(\theta, t) = \frac{s(t)}{(1 - s(t))}H_1 + (\theta + \theta_c)H_2. \quad (18)$$

Comparing with Eq. (7), the gap closing for this Hamiltonian occurs at $s_c = (\theta + \theta_c)/(\theta + 2\theta_c)$, which is $\geq 1/2$. At $t = 0$, $s$ is taken to be 0 as before and the initial state is the ground state of $H_2$, which can be easily prepared. Using the same adiabatic evolution as before to keep the system in the instantaneous ground state, $s$ is then increased until the value $s = 1/2$ is reached. As the gap closing point is not explicitly crossed, the system always stays on one side of the criticality. This prevents the unwanted creation of excitations that would result in fidelity reduction. At $s = 1/2$, the Hamiltonian given by Eq. (18) takes the form of Eq. (7), and the created probe state is used to estimate $(\theta + \theta_c)$. To obtain the unknown parameter $\theta$, one needs to subtract $\theta_c$ from the estimated value. Numerical confirmation of this procedure is displayed in Fig. 5 with the Grover model on a 20-qubit system near the critical point $\theta_c = 1$. As Fig. 5a shows, the fidelity of the prepared state with the actual ground state stays very close to unity. Consequently the QFI and the CFI calculated with the true ground state and the prepared state also match, as shown in Fig. 5b and c, respectively.

We also note that such a time-dependent preparation scheme follows the QFI bound in Eq. (6) (see the Methods section). Additionally, using Eq. (5) we can now find a bound for the rescaled QFI as a new figure of merit. Since the time needed to prepare the ground state is $T \sim 1/\Delta_c \geq 1/\Delta$, one can easily show that the rescaled QFI is bounded as $F^Q/T \leq \frac{\|\partial_\theta H_\theta\|^2}{\Delta}$. This indicates that even when time is incorporated in our resource analysis, the rescaled QFI still benefits from the scaling of both the energy gap as well as the $\|\partial_\theta H_\theta\|^2$. In all the above examples, the exponential advantage comes from the energy gap. Indeed, the

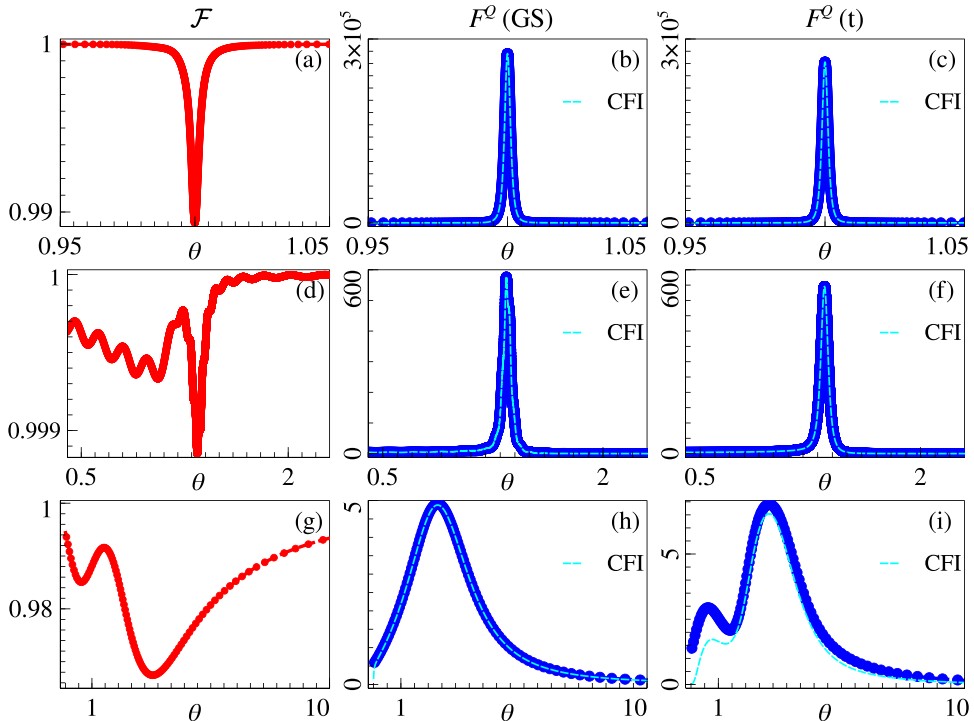

**Fig. 4 | Adiabatic state preparation.** (Top row) Grover model. **a** Fidelity $\mathcal{F}$ of the adiabatically evolved state with the instantaneous ground state. **b** QFI and CFI of the instantaneous ground state. **c** QFI and CFI of the adiabatically evolved state. (Middle row) $p$-spin model. **d** Fidelity $\mathcal{F}$ of the adiabatically evolved state with the instantaneous ground state. **e** QFI and CFI of the instantaneous ground state. **f** QFI and CFI of the adiabatically evolved state. (Bottom row) biclique spin system. **g** Fidelity $\mathcal{F}$ of the adiabatically evolved state with the instantaneous ground state. **h** QFI and CFI of the instantaneous ground state. **i** QFI and CFI of the adiabatically evolved state. 20-qubit system was used for the Grover model, 30 qubits for the $p$-spin model, and a 5-qubit system with $J = 1$, $W_A = 4J$ and $W_B = 3.5J$ was used for the biclique system.

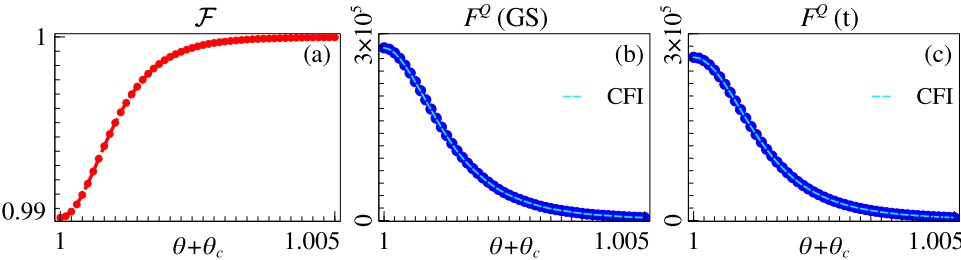

**Fig. 5 | Probe state preparation with unknown parameter.** **a** Fidelity of the adiabatically prepared state with actual ground state. QFI and CFI of (**b**) the ground state and (**c**) the adiabatically prepared state. The results are shown for the Grover model with 20-qubits.

dependence of the bound of the rescaled QFI on the energy gap indicates the exponential advantage even after considering time as a resource.

### Optimal basis

As shown in Fig. 4, it is possible to determine a set of measurement basis relevant for experimental realization, that seem to be optimal. For the Grover model, $\{|m\rangle, |m^\perp\rangle\}$ is an optimal basis. For the $p$-spin model, the total magnetization is one optimal basis. For the biclique spin system, the imbalance between the total magnetization in the two subsystems is given by the operator $\mathcal{I} = \sum_{j_A=1}^{L_A} \sigma_{j_A}^z - \sum_{j_B=1}^{L_B} \sigma_{j_B}^z$. The eigenbasis of this operator serves as an optimal basis.

### Decoherence

Dephasing is a common source of decoherence in spin system dynamics. To quantify the robustness against dephasing during adiabatic evolution, we employ the master equation formalism for the system density operator $\rho$,

$$\dot{\rho} = -\frac{i}{\hbar}[H, \rho] + \frac{\gamma}{2}\sum_n (2c_n \rho c_n^\dagger - c_n^\dagger c_n \rho - \rho c_n^\dagger c_n), \qquad (19)$$

where $\gamma$ is the effective rate of decoherence and $c_n$ is the Lindblad operator. For the Grover model, $H = H_{\text{Grover}}$ and there is only one Lindblad operator $\sigma^z$ between the states $|m\rangle$ and $|m^\perp\rangle$. Our calculations show that even up to a strong decoherence strength $\gamma = 0.1$, the signatures of first order phase transition remain intact along with the exponential growth of critical QFI (see Fig. 6a). Moreover, $F_c^Q$ shows an algebraic decay with increasing decoherence strength (see Fig. 6b for 30 qubits with exponent $\approx 0.93$).

For $p$-spin model, $H = H_{p\text{-spin}}$ and the Lindblad operators are $\sigma_j^z$. Our calculations show that the exponential growth of $F_c^Q$ is retained in this case as well, although up to a lower decoherence strength $\gamma = 0.01J$

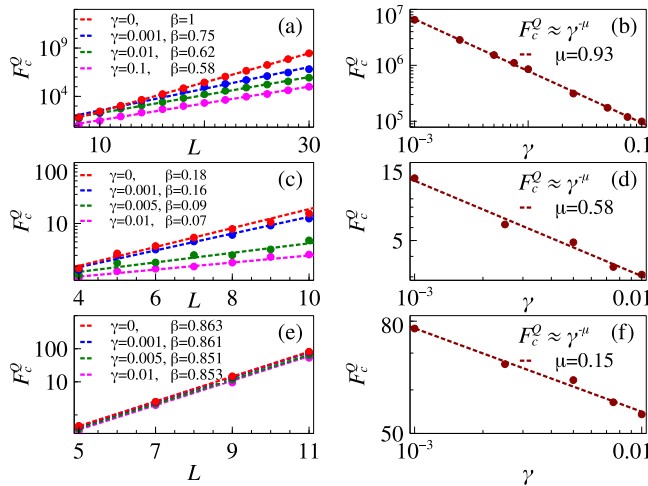

**Fig. 6 | Dephasing dynamics.** (Top row) Grover model with: (**a**) scaling of critical QFI $F_c^Q$ for various decoherence strength $\gamma$; and (**b**) $F_c^Q$ as a function of $\gamma$ at a fixed system size $L = 30$. (Middle row) $p$-spin model with: (**c**) scaling of $F_c^Q$ for various $\gamma$; and (**d**) $F_c^Q$ as a function of $\gamma$ at $L = 10$. (Bottom row) biclique spin system ($J = 1$, $W_A = 4J$, and $W_B = 3.5J$) with: (**e**) scaling of $F_c^Q$ for various $\gamma$; and (**f**) $F_c^Q$ as a function of $\gamma$ at $L = 11$.

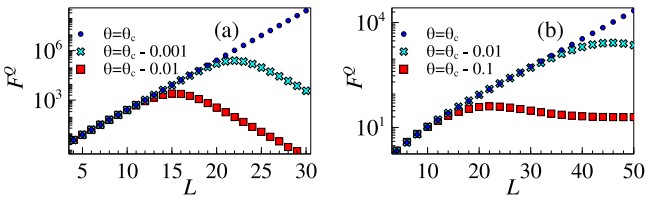

**Fig. 7 | QFI away from criticality. a** Grover model. **b** $p$-spin model. The maximum size $L$ to sustain exponential sensitivity increases with decreasing distance from criticality. Our adaptive sensing strategy utilizes this feature.

(see Fig. 6c). For the algebraic decay of $F_c^Q$ with increasing decoherence strength for 10 qubits, the exponent was $\approx 0.58$ (see Fig. 6d).

For the biclique system with same local Lindblad operators $\sigma_j^z$, we also found that the exponential growth of $F_c^Q$ is retained up to a lower decoherence strength $\gamma = 0.01J$ (see Fig. 6e). Up to this strength we see the effect of decoherence is quite weak on the critical QFI values. For the algebraic decay of $F_c^Q$ with increasing decoherence strength for 11 qubits, the exponent was $\approx 0.15$ (see Fig. 6f).

## Implementation

Realizing the Grover model requires all-to-all connectivity that can be provided by strongly coupled cavity modes[59–61]. Such connectivity would also be useful for $p$-spin models. However, another connection between $p$-spin model and ultra-cold bosons bouncing on an oscillating atom mirror was established in ref. 62. The dynamics can be described effectively by a two-mode Bose-Hubbard model when the driving frequency of the mirror is twice of the natural frequency of the bosons falling onto the mirror under gravity[63]. Mapping between the bosonic operators and spin operators leads to the realization of $p$-spin models with $p = 2$ for two-body contact interaction. Higher order interactions are speculated to give rise to higher $p$-spin models that are considered in this work.

The biclique system can in principle be implemented in the D-Wave Pegasus or Zephyr architecture. Although, due to limited coherence time, schedule control, and constrained measurement processes, the merit of the near-term experiments might be limited. Specifically, the

Pegasus graph of the *D-Wave Advantage system5.4* device hosts 5614-qubits among which one can find the correct embedding of the biclique graph in the setup by using the D-Wave Ocean python package. The architecture already contains 8-qubit Chimera cells with complete bipartite connectivity[64], that can be further coupled by external couplers to achieve a maximum connectivity of 1 qubit to 15 qubits. Thus, the maximum system size of the biclique model that can be simulated in D-Wave architecture is $L = 29$. One has to then initialize the system by setting up the local fields $h_A$ and $h_B$ in the positions of the real qubits and the couplings $J$. Finally, using the standard quantum annealing protocols to tune $\theta$, one may observe the exponentially enhanced sensitivity near the critical points described in this work.

## Adaptive estimation strategy

In criticality-based sensing strategies, the quantum advantage is dominantly available in the vicinity of the critical point. Therefore, one needs to tune the probe, e.g. by applying an external control field, to operate near criticality and achieve the best performance. Away from criticality, the scaling advantage is typically available up to a finite system size. In Fig. 7a, the QFI $F^Q(\theta)$ for the Grover model is plotted against system size $L$ for different distances $\delta$ from criticality. This shows how the optimal length increases with decreasing $\delta$ and the maximum QFI value achievable increases exponentially. The results are qualitatively same for the $p$-spin model, as shown in Fig. 7b. Although the biclique system shows similar trends, due to the limitation of small system sizes we do not include it in this report. Based on the behaviour of the optimal lengths, an adaptive strategy is needed to obtain and update prior information iteratively about the unknown parameter[65–68].

We now exemplify this adaptive strategy with the Grover model for which analytical results are available. From the expression of QFI in Eq. (15), it is easy to see that for any $\delta$ departure from criticality, i.e. $\theta = \theta_c \pm \delta$, QFI maximizes for a system size $L_\delta$, see Fig. 7a. Using a probe with size $L_\delta$ one can reach a precision which, at the worst case, is determined by the minimum QFI attained in the range $[\theta_c - \delta, \theta_c + \delta]$. This helps us to track the maximum uncertainty. It is easy to see that this quantity is $F_{\min}^Q = F^Q(L_\delta, \theta_c + \delta) = \frac{1}{\delta^2(2+\delta)^2}$, with the corresponding optimal system size

$$L_\delta = \log_2\left(\frac{2\delta(\delta+2)+4}{\delta^2}\right). \tag{20}$$

The adaptive strategy can now be summarized in terms of a two-step process within each iteration:

- At the $n$-th step, we assume that we have a prior knowledge about the unknown parameter as $\theta_{est}^{(n-1)} \pm \delta^{(n-1)}$, where $\delta^{(n-1)}$ is the uncertainty of our knowledge. Then, based on this prior knowledge, a control field $\theta_{ctl}^{(n)}$ is applied such that the total effective parameter is $\theta_{est}^{(n-1)} + \theta_{ctl}^{(n)} = \theta_c$. For the given uncertainty $\delta^{(n-1)}$, one can select a probe for this step with optimal size $L^{(n)} = L_{\delta^{(n-1)}}$, see Eq. (20). For this probe size, a single use of the probe takes time $T^{(n)} \sim 1/\Delta^{(n)}$, where $\Delta^{(n)}$ is the energy gap of the probe of size $L^{(n)}$.
- With this probe we perform $M$ measurements, which requires the time resource of $MT^{(n)}$ at this iteration, to update the estimation of the effective parameter to $\theta_{eff}^{(n)}$ with a better precision $\delta^{(n)}$. By deducting the control field, the new prior information for the next step is obtained as $\theta_{est}^{(n)} \pm \delta^{(n)}$. It is worth emphasizing that the uncertainty $\delta^{(n)}$ will be used for choosing the probe size in the next iteration. Note that as the precision is improved, the optimal probe size gets larger which in turn further improves the precision.

These steps are repeated until the desired precision is achieved.

Now we show explicitly how the uncertainty $\delta^{(n)}$ is improved iteratively. Assuming that our sample size $M$ is large enough and the estimator is optimal, one can saturate the Cramér-Rao bound. As we want to ensure that even the maximum possible error is improved

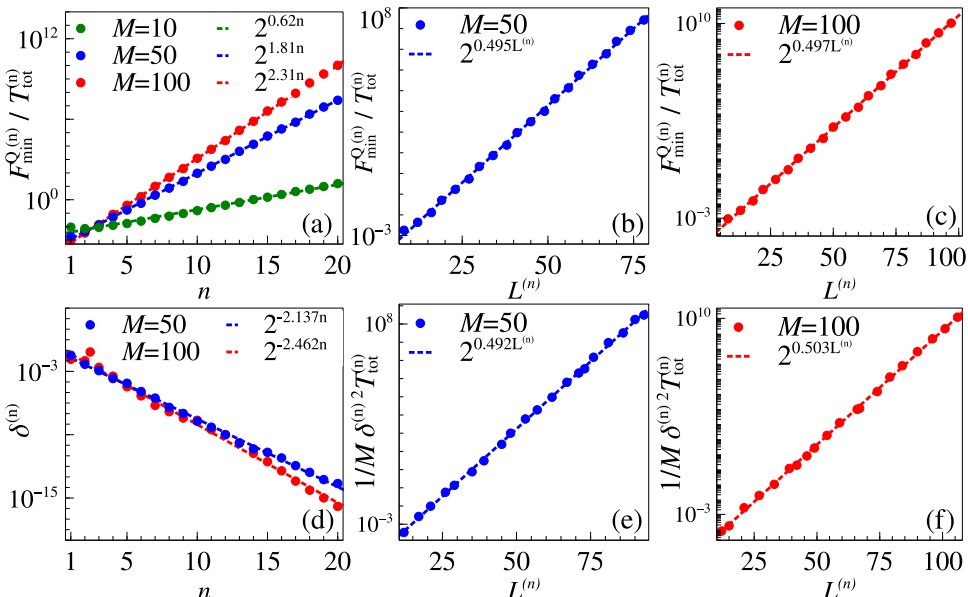

**Fig. 8 | Resource analysis of the adaptive strategy.** (Top row) Maximum uncertainty with optimal estimator with initial uncertainty 0.1. The ratio of QFI and cumulative preparation time at each iteration vs. (**a**) iteration step, and vs. system size at each step with measurement numbers (**b**) $M = 50$, (**c**) $M = 100$. (Bottom row) Uncertainty with Bayesian estimator with true parameter 0.9 and initial range [0.8, 1.1]. **d** Iterative uncertainty at each step. Iterative figure of merit vs. system size at each step with (**e**) $M = 50$, (**f**) $M = 100$.

iteratively, we consider the worst scenario at each step where the true parameter value is the farthest from the critical point. Here, the uncertainty becomes $\delta^{(n)} \simeq 1/\sqrt{MF_{\min}^{Q(n)}} = 1/\sqrt{MF^Q(L^{(n)}, \theta_c + \delta^{(n-1)})}$. To evaluate the performance of the probe after $n$ iterations, while incorporating the total time as the resource, one has to consider the figure of merit as $F_{\min}^{Q(n)}/T_{\text{tot}}^{(n)}$, where $T_{\text{tot}}^{(n)} = M\sum_{k \le n} T^{(k)}$ is the total time spent to reach this stage. Now, by inverting Eq. (20) to get $\delta^{(n)}$ in terms of $L^{(n)}$ and incorporating it in Eq. (15) we get, $F_{\min}^{Q(n)} \sim 2^{L^{(n)}}$ and $T_{\text{tot}}^{(n)} \le MnT^{(n)} \sim Mn/\Delta^{(n)}$. As $\Delta^{(n)}$ is lower bounded by its critical value $\Delta_c^{(n)}$, we recall the scaling relations for the Grover model from Eqs. (14)–(15), and write

$$\frac{F_{\min}^{Q(n)}}{T_{\text{tot}}^{(n)}} \ge \frac{F_{\min}^{Q(n)}}{Mn/\Delta_c^{(n)}} \sim \frac{2^{L^{(n)}/2}}{Mn}. \tag{21}$$

This clearly shows that the adaptive rescaled QFI scales exponentially with the probe size $L^{(n)}$. Nonetheless, we still numerically investigate the scaling of $F_{\min}^{Q(n)}/T_{\text{tot}}^{(n)}$ with respect to both $n$ and $L^{(n)}$. In Fig. 8a, we see that $F_{\min}^{Q(n)}/T_{\text{tot}}^{(n)}$ falls off exponentially with step number $n$, which signals that the adaptive strategy is very efficient even with few iterative steps and very modest number of measurements $M$. In Fig. 8b and c, we show that the exponential scaling of $F_{\min}^{Q(n)}/T_{\text{tot}}^{(n)}$ is indeed retained as was predicted above. Therefore we conclude that even with the consideration of the largest uncertainty at each step with finite $M$ measurements, while accounting for all the resources, the exponential scaling advantage is retained in this adaptive strategy.

The above analysis assumes that the Cramér-Rao bound is achievable at all the iterations using $M$ measurements. Now, we show that this is indeed possible by performing a Bayesian estimation[69,70], while keeping $M$ to be a modest value. As discussed before, at the $n$-th step of the iterative procedure, we apply a control field $\theta_{\text{ctl}}^{(n)}$, based on our previous estimation $\theta_{\text{est}}^{(n-1)} \pm \delta^{(n-1)}$, to make sure that the probe operates around the critical point. We prepare the probe in its ground state $|GS\rangle$ corresponding to the total effective parameter. We choose the measurement basis $\{\Pi_k\}$ as the one specified in the 'Optimal basis'

section. We then simulate generating $M$ number of experimental data by randomly sampling from the probability distribution of the ground state in this basis. If the $k$-th outcome is obtained $n_k$ times, then $\sum_{k=1}^d n_k = M$ and $d$ is the total number of possible outcomes. For the Grover model that we consider here, $d = 2$. This measured probability distribution $\{n_k/M\}$ is then compared with the model probability distribution $\{p_k = \langle GS|\Pi_k|GS\rangle_{\theta^{(n)}}\}$. This is done with the aid of the 'likelihood' function, given by the multinomial distribution $P(\{n_k\}|\theta) = \frac{M!}{\prod_k n_k!}\prod_k p_k^{n_k}$. If no information is available other than the range of $\theta^{(n)}$ between $\theta_{\min}$ and $\theta_{\max}$, then the initial 'prior' is the uniform distribution $P(\theta) = \frac{1}{\theta_{\max} - \theta_{\min}}$. Using Bayes' theorem, we now write the 'posterior' distribution $P(\theta|\{n_i\}) = P(\{n_i\}|\theta) P(\theta)$, and normalize it. In this work, at each iteration $n$, we take the prior to be a flat distribution in the range $[\theta_{\text{est}}^{(n-1)} - 5\delta^{(n-1)}, \theta_{\text{est}}^{(n-1)} + 5\delta^{(n-1)}]$. Note that one can also use the posterior of the $(n-1)$-th iteration as the prior, however, this can cause large fluctuations that would demand large $M$ to converge. For large enough $M$, the final posterior distribution is Gaussian, the mean and standard deviation of which serve as the $\theta_{\text{est}}^{(n)}$ and $\delta^{(n)}$, respectively. Although $M$ is typically a few thousands in experiments, here $M = 50$ or $M = 100$ was sufficient. As shown in Fig. 8d, the uncertainty at each step $\delta^{(n)}$ falls off exponentially with $n$ and the exponents grow in magnitude as $M$ is increased. For the purpose of resource analysis, we then look at $1/M\delta^{(n)2}T_{\text{tot}}^{(n)}$, which is an analogue of the ratio of QFI and total preparation time that was considered before. As shown in Fig. 8e and f, not only does this quantity show the desired exponential scaling, it is also quantitatively similar to the case of optimal estimators shown in Fig. 8b and c. This empirical analysis based on Bayesian estimation clearly demonstrates that the adaptive strategy is very effective to harness the exponential advantage even if the unknown parameter of interest is away from the critical point.

## Discussion

To utilize quantum features for enhancing sensing precision several strategies have been put forward which resulted in sensors based on GHZ-like entangled states, criticality and non-equilibrium dynamics. In

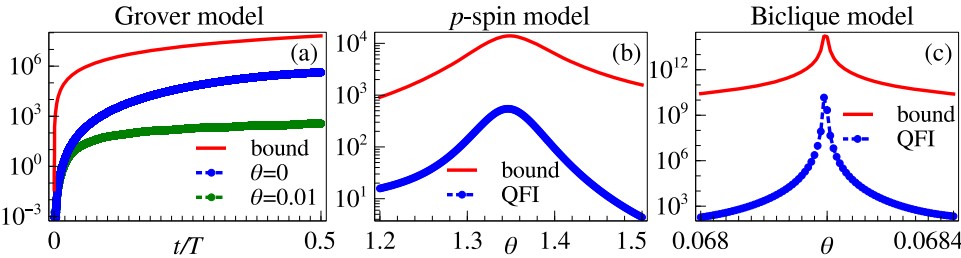

**Fig. 9 | Upper bounds of QFI. a** QFI evolution within the bound during ground state preparation of the Grover model with 20 qubits. Upper bounds and QFI of the ground states in (**b**) $p$-spin model with 30 spins, and (**c**) the biclique model with 9 spins.

most of these methods, the QFI scales algebraically with respect to system size, i.e. $F^Q \sim L^\beta$. Surpassing algebraic advantage and reaching exponential scaling has remained elusive when all the resources, such as the preparation time, are taken into account. Here, we have shown that a class of systems with first order quantum phase transitions with exponential energy gap closing can indeed achieve exponential scaling for the QFI. Remarkably, the exponential scaling nature is preserved even if the state preparation time, through local adiabatic driving, is accounted for. We have illustrated our results by considering three distinct models, namely Grover, $p$-spin, and biclique spin systems, featuring first order phase transition. The results comply with the fundamental bounds that have been established for quantum probes. In addition, they are robust against moderate decoherence and the optimal bases are also experimentally realizable. While criticality-based sensing is inherently local in nature, we have shown, with an adaptive estimation strategy, that it is always possible to harness the exponential scaling for sensing arbitrary parameters to unprecedented precision. Our results can in principle be verified with D-Wave quantum devices in which the biclique spin system may be implemented. This work paves the way for a concrete precision sensing strategy with applications in estimating fundamental physical constants, which require ultra-accurate local probes.

## Methods
### QFI bounds
We first show that the ground state QFI for the Grover model in Eq. (15) is upper bounded according to Eq. (5) by $\|H_2\|^2/\Delta^2 = N^2/(N^2(1-\theta)^2 + 4N\theta)$. To see this, we start with

$$(N(\theta - 1) + 2)^2 \geq 0$$
$$\Rightarrow N^2(1-\theta)^2 + 4N\theta \geq 4(N-1)$$
$$\Rightarrow N \geq \frac{4(N-1)}{[N(1-\theta)^2 + 4\theta]}$$
$$\Rightarrow \frac{N^2}{[N^2(1-\theta)^2 + 4N\theta]} \geq \frac{4(N-1)}{[N(1-\theta)^2 + 4\theta]^2}, \tag{22}$$

which proves the desired relation as the LHS is the upper bound and the RHS is the QFI. We also notice that at criticality ($\theta = 1$), the QFI almost saturates the upper bound for large system sizes. Additionally, for the ground state preparation scheme presented in the main text according to the evolution under the Hamiltonian in Eq. (18), the relevant bound for the time-dependent QFI is given by Eq. (6). In Fig. 9a we show that this bound is satisfied during state preparation both at criticality and away from it. The upper bounds and the QFI of the ground states for the $p$-spin and biclique models near criticality are shown in Fig. 9b and c, respectively.

### Preparation time
For the adiabatic state preparation based on the Eq. (17), the condition for the fidelity of the evolved state with the instantaneous ground state

to be large, namely, $\mathcal{F}(t) = |\langle GS(t)|\psi(t)\rangle|^2 \geq 1 - \epsilon^2$, is

$$\frac{|\langle\psi_1(t)|\frac{d}{dt}H(t)|GS(t)\rangle|}{\Delta(t)^2} \leq \epsilon, \tag{23}$$

with $|\psi_1(t)\rangle$ as the instantaneous first excited state. Transferring the time-dependence on $s(t)$, we can write

$$\frac{dt}{ds} \geq \frac{|\langle\psi_1(s)|\frac{d}{ds}H(s)|GS(s)\rangle|}{\epsilon\Delta(s)^2}. \tag{24}$$

For the $p$-spin model, we numerically observe that $|\langle\psi_1(s)|\frac{d}{ds}H(s)|GS(s)\rangle| < L$. Therefore we take the preparation time for the ground state at $s$,

$$T(s) = \int_0^s \frac{L}{\epsilon\Delta(s)^2} ds. \tag{25}$$

The resulting time was found to scale as $\sim e^{0.055L}$ for $p = 3$ and $\lambda = 1$, which is advantageous as this exponent as even smaller than that of $1/\Delta_c$. Similar results were found for the biclique system as well.

## Data availability
The datasets generated during and/or analysed during the current study are available in the Github repository https://github.com/SaubhikSarkar/QFI_First_Order_Phase_Transition.

## Code availability
The code used in this study is available at the Github repository https://github.com/SaubhikSarkar/QFI_First_Order_Phase_Transition.

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

## Acknowledgements

A.B. acknowledges support from National Natural Science Foundation of China (Grants Nos. 12050410253, 92065115, and 12274059) and the Ministry of Science and Technology of China (Grant No. QNJ2021167001L). S.S. acknowledges support from National Natural Science Foundation of China (Grant No. W2433012). R.G. and S.B. acknowledge EPSRC grants EP/Y004590/1 MACON-QC and EP/R029075/1 Non-Ergodic Quantum Manipulation for support.

## Author contributions

A.B. and S.S. proposed the resource for quantum-enhanced sensitivity while R.G. and S.B. proposed the concrete systems for implementation. R.G. performed the analytical calculations and S.S. performed the numerical simulations. All the authors contributed in writing the manuscript.

## Competing interests

The authors declare no competing interests.
