## [Transparent Peer Review file · Nature Communications]

Exponentially-enhanced quantum sensing with many-body phase transitions

Corresponding Author: Dr Saubhik Sarkar

Version 0:

Reviewer comments:

Reviewer #1

(Remarks to the Author)

The Authors show that exponential scaling in quantum metrology using first order phase transitions is achievable even when the time needed to create the initial state is considered. They also show that this result is robust against moderate dephasing.

I believe the paper is nice and well written. The results are very interesting and relevant. I believe the paper deserves publication.

I just have a simple request for clarification. Typically, these super-Heisenberg scalings are useless because, while it's true that the QFI has a super-Heisenberg scaling, this happens only for an extremely limited interval of the parameter. Namely, one has to already know the value of the parameter to a very high precision in order to tune the estimation strategy to make use of the super-Heisenberg scaling. But this makes the whole estimation useless: it requires one to already know the value of the parameter to such high accuracy that it is not even necessary to estimate it. In other words, if one takes into account the prior information that is required on the parameter in order to implement the estimation, then such prior info by itself allows one to know the value of the parameter up to the precision that the estimation itself achieves: this makes the estimation useless. In this specific case, the Authors seem to get around this issue by using an adaptive estimation strategy detailed in the section with the eponymous title. However, it is not clear to me that the exponential scaling is retained also when one counts all the steps of the adaptive strategies. It seems to me that the adaptive error epsilon will decrease too slowly. I'm probably mistaken, so it would be nice if the Authors could clarify that.

In essence I definitely endorse the publication of this paper, once the Authors have satisfactorily replied to the above request for clarification.

(Remarks on code availability)

Reviewer #2

(Remarks to the Author)

The draft by Sarkar et al. considers some prototypical quantum many-body systems exhibiting the first-order phase transition. Performing mostly small-scale exact diagonalization studies, the authors argue that this class of systems offers exponentially enhanced sensing opportunities, even when full resource analysis is performed.

The article has two fatal weaknesses:

- 1) I have not learned anything fundamentally new from this draft that I did not already know about.
- 2) The core claim that the Quantum Fisher Information "(QFI) scales exponentially even after taking preparation time into account" is clearly false. It is based on elementary misunderstandings on the author's part.

Let me expand on those points. The relation between how quickly the (ground) state changes in the vicinity of the quantum critical point (potential usefulness to sensing is all about that) and how the minimal energy gap behaves, and consequently

what the order of the quantum phase transition is, are well established and understood. Minimal examples can be seen directly by considering 2-level avoided level crossing systems (with the Hamiltonian in Eq. (9) being an example) and performing trivial diagonalization of such a two-by-two matrix. This is a textbook knowledge. I could recommend the author's book "Elements of Phase Transitions and Critical Phenomena" by Nishimori and Ortiz as one of many places where the authors can read about it. This makes it hard for me to understand making statements like the one appearing in the introduction: "This gives rise to the conjecture that the energy gap closing might be the reason behind quantum-enhanced sensing [38]".

Regarding the core claim of the paper. It is long established that the bound on the improvement of the measurement goes linearly in time. Equivalently, the bound on QFI grows quadratically with time. This is the element of the Heisenberg limit the authors write about in the first paragraph, citing [6-8]. Ref [9], which the authors refer to in the middle of the first paragraph, gives a generally applicable mathematical framework showing this. It is important to note that all the examples discussed by the authors fall within that framework when the preparation time is included. The authors also clearly know this as they write that "QFI also depends on evolution-time t and typically scales at t^2 ..." (see the end of the first paragraph).

That is why, considering QFI divided by time, see the text below Eq. (14), and claiming that the fact it grows exponentially with system size (and evolution time) is meaningless. The correct quantity to look at here is QFI divided by time squared, and this behaves exactly as expected.

To summarize, there is nothing special here and the first-order phase transition does not offer any exponential-enhanced sensing when preparation time is taken into account. The authors have exponentially long preparation times and claim exponential sensing based on this. This is nothing new or promising.

If the current draft shows anything, it is that the first-order phase transitions are exponentially bad for sensing -- exactly the opposite of what the authors claim. Indeed, the high-sensitive critical point gets exponentially localized around the transition point as the system size increases. This calls for exponential precision needed in controlling and knowing the physical system, which is unrealistic. All the above is additionally based on a fine-tuned preparation protocol that again requires precise knowledge about the state of the system. This does not leave any room for sensing. With a standard driving protocol that relaxed some of those assumptions, one can expect much worse performance (see an element of the discussion above Eq. (14) on the adiabatic evolution times), and likely very far from the Heisenberg limit.

With the above general remarks, I don't believe there is any need for more specific comments though I have to say the draft is generally written sloppily. The authors, for instance, mix upper bounds with lower bounds, which is something that should have been eliminated even after a casual reading.

Having said that, I still have to comment on the implementation section. The authors suggest using DWave quantum annealing for a biclique model to realize their "sensing proposal". However, DWave annealer is a quantum simulator. Let's assume for a second, we have access to an ideal machine. Namely, the parameters of the simulator are set by the user. I cannot imagine what kind of sensing the authors have in mind in such a scenario where they set the precise parameters of the machine. This fundamentally does not make any sense, and it does not even take into account the fact that the actual machine is noisy, the level of control is highly limited, quench protocols it offers are fixed and far from the ideal assumed by the authors of that draft, and finally the annealer does not allow for measurement at arbitrary point.

(Remarks on code availability)

Reviewer #3

(Remarks to the Author)

This work is an interesting contribution to the active field of quantum critical metrology. It shows that many-body interactions can be harnessed to generate a scaling of the measurement precision, quantified by the quantum Fisher Information (QFI), that grows exponentially with the system's size. This remains true when the figure of merit is the QFI rate, and also under mild decoherence and imperfections. This main result is obtained by carefully analyzing a very good selection of insightful examples, ranging from a simple analytically solvable Groover model to more realistic spin systems. Explicit adiabatic protocols as well as implementations are also developed. Furthermore, the paper is also very well written and clear to read.

Overall, I consider this a significant contribution to the field of quantum metrology, which may fit well in Nat. Comm. Having said that, I have a main comment/criticism that should be addressed before publication.

Basically, from reading the paper it does not become clear whether critical metrology, and in particular the protocols considered, should obey any limit in terms of precision. Because of the exponential increase of the QFI with N , it also gives the impression that the Heisenberg limit is violated. This is however clearly not true, and the paper should clarify it.

First, when time is taken into account (as the authors do in the adiabatic protocols), the QFI should always be smaller than $t^2 \|H_2\|^2$, where H_2 is from Eq. (5), see [Phys. Rev. X 8, 021022 (2018), Ref. [21] of the paper] as well as [PRL 98, 090401 (2007), Ref. [9] of the paper]. This is an important point to clarify: despite the exponential growth, all protocols satisfy the (generalized) Heisenberg limit.

Second, when time is not considered a resource, the fundamental limits of the QFI of ground-state metrology have recently been established in [PRL 134, 010801 (2025)], where the QFI has been bounded by $\|H_2\|^2/\Delta^2$ (see Eq. (18)). The present results should also satisfy this bound, as the exponential growth of the QFI arises due to the exponential decay of the gap. This also agrees well with the intuition of the authors that the QFI diverges because the gap closes.

Summarizing, despite the exponential growth, the QFI is still bounded by physical limits: both the generalized Heisenberg limit (when time is a resource) and by the recent QFI bound of PRL 134, 010801 (2025). The paper would gain in clarity if the protocols are contrasted with such bounds, and it is explained why and in which precise sense the Heisenberg limit is not violated.

(Remarks on code availability)

Version 1:

Reviewer comments:

Reviewer #1

(Remarks to the Author)

I believe the Authors have satisfactorily replied to my concerns and to those of the other referees. I think the observation of Referee 2 that the exponential enhancement effectively comes from an exponential increase in the time is an important one for the understanding of the result. While the Authors seem to agree with it, it might be worth adding a sentence of this type in the introduction together with the fact that the current result still satisfies the t^2 bound. I think this would be a very simple, yet important, addition to the paper to avoid future miscomprehensions.

(Remarks on code availability)

Reviewer #2

(Remarks to the Author)

I carefully read the revised version. The authors did a good job filling-in many holes in the discussion. I am still considering such a protocol as not very realistic, as exponential precision is stemming from exponentially-long evolution times (the authors use this very argument in paragraph 2 to argue against some non-Hermitian systems proposals), where a lot of control parameters appear to require similar degree of control for the proposal to be successful.

I'm however willing to accept the argument and the proposal.

I'm still finding D-Wave proposal as overly optimistic. The authors assume long coherent evolution time, control of the driving schedule, possibility to measure at arbitrary point in the quench. None of this is available in the actual machine, while the authors make it sound as if such an experiment could be performed in the available platform (which is not true).

Minor comments:

$|\langle GS(t) | \Psi(t) \rangle|^2 < 1 - \epsilon^2$. still seem to have incorrect direction; $1 - \epsilon^2$ should be a lower bound, not an upper bound if one believes in Eq. (5) of Ref [57]. Ditto for the same inequality in other places in the text.

(Remarks on code availability)

Reviewer #3

(Remarks to the Author)

I am satisfied with the response to my questions and the changes in the manuscript, which has considerably improved after revisions. I recommend acceptance.

As a minor remark, I would suggest introducing (7) before (5)-(6), so that the bounds can be expressed as a function of H_2 (avoiding introducing two different notations for the same quantity).

(Remarks on code availability)

Manuscript NCOMMS-24-69490

Exponentially-enhanced quantum sensing with many-body phase transitions

by S. Sarkar, A. Bayat, S. Bose, and R. Ghosh

Point-by-point response to the reviewers' comments

We thank the reviewers for reading our manuscript and for the insightful questions, criticisms, and suggestions. In this document, we provide point by point response to the comments of the three reviewers. We are very glad that Reviewer 1 deems that “the paper deserves publication”, subject to a question on the adaptive strategy. We are also delighted to see that Reviewer 3 finds our paper “a significant contribution to the field of quantum metrology, which may fit well in Nat. Comm.”, subject to comparison with the fundamental bounds. We have addressed these issues diligently and as a result, the revised manuscript has significantly improved in quality. On the other hand, Reviewer 2 has criticized the paper on the basis of incorrectly assessing our resource analysis. In our detailed responses to the comments, we have explained why the arguments of Reviewer 2 are wrong and provided evidence from literature regarding the validity of our analysis.

The summary of all the changes is the following:

- A new section named “Fundamental QFI bounds in many-body probes” has been added.
- The section “Adaptive estimation strategy” has been entirely rewritten.
- A new section named “QFI bounds” has been added in the Methods.
- Two new figures, namely Fig. 8 and Fig. 9, have been added.
- Several new references have been added.
- The whole paper has been revised according to the comments.
- The codebase has also been updated to account for the new calculations.

We hope that, after addressing all the comments by the reviewers while providing the corresponding justifications, the revised manuscript will be found appropriate for publication in Nature Communications.

Responses to the report of reviewer 1

Reviewer Comment: The Authors show that exponential scaling in quantum metrology using first order phase transitions is achievable even when the time needed to create the initial state is considered. They also show that this result is robust against moderate dephasing.

I believe the paper is nice and well written. The results are very interesting and relevant. I believe the paper deserves publication.

Response: We thank the reviewer for reading our manuscript and for the positive assessment.

Reviewer Comment: I just have a simple request for clarification. Typically, these super-Heisenberg scalings are useless because, while it's true that the QFI has a super-Heisenberg scaling, this happens only for an extremely limited interval of the parameter. Namely, one has to already know the value of the parameter to a very high precision in order to tune the estimation strategy to make use of the super-Heisenberg scaling. But this makes the whole estimation useless: it requires one to already know the value of the parameter to such high accuracy that it is not even necessary to estimate it. In other words, if one takes into account the prior information that is required on the parameter in order to implement the estimation, then such prior info by itself allows one to know the value of the parameter up to

the precision that the estimation itself achieves: this makes the estimation useless. In this specific case, the Authors seem to get around this issue by using an adaptive estimation strategy detailed in the section with the eponymous title. However, it is not clear to me that the exponential scaling is retained also when one counts all the steps of the adaptive strategies. It seems to me that the adaptive error epsilon will decrease too slowly. I'm probably mistaken, so it would be nice if the Authors could clarify that.

Response: We thank the reviewer for raising an important issue. Indeed, similar to all the criticality-based quantum sensors, our protocol fits within the context of *local sensing*, where some prior information about the unknown parameter is needed. However, deviations from the critical point can iteratively be improved so that the probe quickly operates at around its critical point, where its precision is the best and scales exponentially with the probe size. To elaborate on this, we have entirely rewritten our section “Adaptive estimation strategy” and performed new simulations with both heuristic analysis, based on quantum Fisher information, as well as fully quantitative Bayesian estimation. The results are displayed in the newly added Fig. 8. As the figure shows, the iterative procedure indeed retains the exponential scaling.

In the iterative procedure, at each step, based on our previous knowledge about the uncertainty δ about the unknown parameter, i.e. $\theta \pm \delta$, we select an optimal probe size L_δ . A single use of this probe at iteration k , requires $T^{(k)}$ time. For any estimation, we can assume that M measurements are performed, making the total time at each iteration $MT^{(k)}$. To determine the scaling of the ratio of QFI and total preparation time, we computed the QFI at n -th step, $F^Q(n)$, and total time $T_{\text{tot}}^{(n)} = M \sum_{k \leq n} T^{(k)}$. By performing this calculation for the Grover model, where analytical results are available, we obtain exponential improvement with respect to both iterations and the probe size. The results are shown in Figs. 8(a)-(c).

To further strengthen our claims, we also exploit Bayesian estimation to explicitly demonstrate the exponential advantage in a practical scenario. The results match well with the heuristic analysis of the quantum Fisher information, as shown in Figs. 8(d)-(f). These results lead us to conclude that even within the adaptive strategy, the exponential scaling is retained.

Reviewer Comment: In essence I definitely endorse the publication of this paper, once the Authors have satisfactorily replied to the above request for clarification.

Response: We thank the reviewer again for supporting our work and raising an important issue. Thanks to the reviewer’s comment, our revised section “Adaptive estimation strategy” is significantly improved. With the necessary actions taken, we hope that the reviewer deems the revised manuscript fit for publication.

Responses to the report of reviewer 2

Reviewer Comment: The draft by Sarkar et al. considers some prototypical quantum many-body systems exhibiting the first-order phase transition. Performing mostly small-scale exact diagonalization studies, the authors argue that this class of systems offers exponentially enhanced sensing opportunities, even when full resource analysis is performed.

The article has two fatal weaknesses:

- 1) I have not learned anything fundamentally new from this draft that I did not already know about.
- 2) The core claim that the Quantum Fisher Information “(QFI) scales exponentially even after taking preparation time into account” is clearly false. It is based on elementary misunderstandings on the author’s part.

Response: We thank the reviewer for reading our paper. We strongly disagree with the opinions of the reviewer. The first argument is very subjective and is clearly in contradiction with the views of Reviewers 1 and 3. The second comment is wrong as we will elaborate on in the following. The source of this comment is perhaps misunderstanding of the concept of resource in the context of sensing and confusing it with the scaling of the quantum Fisher information, namely $F^Q \sim t^2$ during non-equilibrium dynamics. We will clarify the issue in our responses below and provide previous works in literature to show that our analysis is indeed based on a standardized figure of merit.

Reviewer Comment: Let me expand on those points. The relation between how quickly the (ground) state changes in the vicinity of the quantum critical point (potential usefulness to sensing is all about that) and how the minimal energy gap behaves, and consequently what the order of the quantum phase transition is, are well established and

understood. Minimal examples can be seen directly by considering 2-level avoided level crossing systems (with the Hamiltonian in Eq. (9) being an example) and performing trivial diagonalization of such a two-by-two matrix. This is a textbook knowledge. I could recommend the author’s book “Elements of Phase Transitions and Critical Phenomena” by Nishimori and Ortiz as one of many places where the authors can read about it. This makes it hard for me to understand making statements like the one appearing in the introduction: “This gives rise to the conjecture that the energy gap closing might be the reason behind quantum-enhanced sensing [38]”.

Response: We struggle to understand what exact criticism is being raised by the reviewer in this comment. We guess that the reviewer argues that quantum enhanced sensitivity resulted from gap closing is expected and is not new. Firstly, the statement in question is not a result of this paper but a fact that has been conjectured in the literature. As the recent work (PRL 134, 010801 (2025)) proves, the upper bound of the quantum Fisher information depends both on the energy gap and the seminorm of the relevant part of the Hamiltonian (see Eq. (5) in the main text). To clarify this issue, we have added a new section called “Fundamental QFI bounds in many-body probes” and revised various places in the main text.

Secondly, our contribution in this work is to find certain classes of many-body systems in which quantum Fisher information, and not just its bound, scales exponentially. The exponential advantage remains valid even when we incorporate time as a resource in our analysis. Finding such exponential advantage in different many-body probes is the core novelty of our work. It is worth emphasizing that our findings satisfy the fundamental bounds of the quantum Fisher information, as has been detailed in our responses to Reviewer 3. In fact, complying with fundamental bounds does not degrade our results but provides extra support for their validity.

Reviewer Comment: Regarding the core claim of the paper. It is long established that the bound on the improvement of the measurement goes linearly in time. Equivalently, the bound on QFI grows quadratically with time. This is the element of the Heisenberg limit the authors write about in the first paragraph, citing [6-8]. Ref [9], which the authors refer to in the middle of the first paragraph, gives a generally applicable mathematical framework showing this. It is important to note that all the examples discussed by the authors fall within that framework when the preparation time is included. The authors also clearly know this as they write that “QFI also depends on evolution-time t and typically scales at t^2 ...” (see the end of the first paragraph).

That is why, considering QFI divided by time, see the text below Eq. (14), and claiming that the fact it grows exponentially with system size (and evolution time) is meaningless. The correct quantity to look at here is QFI divided by time squared, and this behaves exactly as expected.

To summarize, there is nothing special here and the first-order phase transition does not offer any exponential-enhanced sensing when preparation time is taken into account. The authors have exponentially long preparation times and claim exponential sensing based on this. This is nothing new or promising.

Response: The reviewer is right about the scaling of the QFI during non-equilibrium dynamics where $F^Q \sim t^2$. This is a well-established fact. However, the reviewer has misunderstood the concept of the resource in quantum sensing schemes. For incorporating time as a resource, we can consider a total time of T_{tot} which has to be spent on sensing. The probe preparation time is T , which in our case is inversely proportional to the energy gap Δ , namely $T \sim 1/\Delta$. Therefore, within the available time resource T_{tot} one can perform $M = T_{\text{tot}}/T$ measurements. By inserting this into the Cramér-Rao inequality one gets

$$\delta\theta \geq \frac{1}{\sqrt{MF^Q}} = \frac{1}{\sqrt{T_{\text{tot}}F^Q/T}}.$$

This analysis clearly shows that when the total time is considered as the resource, the relevant figure of merit would be F^Q/T . In fact, this figure of merit is extensively used in the literature, e.g. see the following papers:

Physical Review Letters 111, 120401 (2013)
 Physical Review X 5, 031010 (2015)
 Physical Review Letters 116, 120801 (2016)
 Quantum 2, 110 (2018)
 Physical Review Letters 125, 200505 (2020).

For better clarification, we have amended the first and the last paragraphs of the “Resource analysis” section.

Reviewer Comment: If the current draft shows anything, it is that the first-order phase transitions are exponentially bad for sensing – exactly the opposite of what the authors claim. Indeed, the high-sensitive critical point gets

exponentially localized around the transition point as the system size increases. This calls for exponential precision needed in controlling and knowing the physical system, which is unrealistic. All the above is additionally based on a fine-tuned preparation protocol that again requires precise knowledge about the state of the system. This does not leave any room for sensing. With a standard driving protocol that relaxed some of those assumptions, one can expect much worse performance (see an element of the discussion above Eq. (14) on the adiabatic evolution times), and likely very far from the Heisenberg limit.

Response: The reviewer has misunderstood the concept of *local sensing*. In this class of sensing problems, which includes sensing with almost all quantum many-body probes, a prior knowledge about the unknown parameter is needed. Therefore, dismissing the sensing capability of many-body probes, including our models which harness first-order phase transitions, is ill-advised. To show how one can achieve exponential sensitivity even for parameters away from criticality in an iterative way, we have significantly revised the section “Adaptive estimation strategy”. This establishes concretely that scaling advantage is retained within the adaptive strategy. We have additionally performed a Bayesian estimation for a practical scenario, which shows the effectiveness of this strategy in reaching very high precision. Therefore, stating that the preparation protocol is “fine-tuned” and “requires precise knowledge about the state of the system” is not correct and one can harness quantum criticality through an adaptive strategy. Lastly, the comment by the reviewer about the standard driving protocol might indeed be useful for sensing, however, it is irrelevant to the subject of this paper.

Reviewer Comment: With the above general remarks, I don’t believe there is any need for more specific comments though I have to say the draft is generally written sloppily. The authors, for instance, mix upper bounds with lower bounds, which is something that should have been eliminated even after a casual reading.

Response: This comment directly contradicts the observations made by the other reviewers. Reviewer 1 has found that “the paper is nice and well written” while Reviewer 3 has commented that the paper is “very well written and clear to read”. Nonetheless, we have done careful proofreading of the paper and tried to fix any existing typos.

Reviewer Comment: Having said that, I still have to comment on the implementation section. The authors suggest using DWave quantum annealing for a biclique model to realize their “sensing proposal”. However, DWave annealer is a quantum simulator. Let’s assume for a second, we have access to an ideal machine. Namely, the parameters of the simulator are set by the user. I cannot imagine what kind of sensing the authors have in mind in such a scenario where they set the precise parameters of the machine. This fundamentally does not make any sense, and it does not even take into account the fact that the actual machine is noisy, the level of control is highly limited, quench protocols it offers are fixed and far from the ideal assumed by the authors of that draft, and finally the annealer does not allow for measurement at arbitrary point.

Response: Quantum sensing is currently at the early stage of its development. Most of the experiments with quantum sensors are indeed proof of principles and far from being actual probes readily suitable for practical applications. In the state-of-the-art experiments to observe quantum-enhanced sensitivity, the parameter to be estimated is typically set by the experimenter. The current objective is to demonstrate that quantum features indeed provide sensing precision beyond the capabilities of the classical sensors while it would be the next generation of experiments that would focus on sensing unknown parameters. We agree with the reviewer that D-Wave platform is a quantum simulator. Nonetheless, it can be used for demonstration of a proof of principle experiment for sensing with the biclique model. Only after confirming with these proof of principles, real quantum devices can be developed for practical applications. Therefore, it is unrealistic to expect that one can sense an unknown parameter using a prototype quantum sensor, including platforms such as the D-Wave architecture.

Responses to the report of reviewer 3

Reviewer Comment: This work is an interesting contribution to the active field of quantum critical metrology. It shows that many-body interactions can be harnessed to generate a scaling of the measurement precision, quantified by the quantum Fisher Information (QFI), that grows exponentially with the system’s size. This remains true when the figure of merit is the QFI rate, and also under mild decoherence and imperfections. This main result is obtained by carefully analyzing a very good selections of insightful examples, ranging from a simple analytically solvable Groover model to more realistic spin systems. Explicit adiabatic protocols as well as implementations are also developed. Furthermore, the paper is also very well written and clear to read.

Overall, I consider this a significant contribution to the field of quantum metrology, which may fit well in Nat. Comm. Having said that, I have a main comment/criticism that should be addressed before publication.

Response: We thank the reviewer for reading our work and for the positive remarks.

Reviewer Comment: Basically, from reading the paper it does not become clear whether critical metrology, and in particular the protocols considered, should obey any limit in terms of precision. Because of the exponential increase of the QFI with N , it also gives the impression that the Heisenberg limit is violated. This is however clearly not true, and the paper should clarify it.

Response: We are thankful to the reviewer for giving us the opportunity to connect our results with the fundamental limits of quantum metrology. Indeed, as we have discussed in the revised manuscript, our results fully fit within the fundamental bounds that have been found for the quantum Fisher information. These bounds even provide useful insight about the results and support the exponential scaling with clarity. In the revised version of the manuscript, we have added an entirely new section “Fundamental QFI bounds in many-body probes” to discuss the relevant bounds to our paper. In addition, for each of the three models we discuss the connection between such fundamental bounds and our results. Our responses to each comment is given below.

Reviewer Comment: First, when time is taken into account (as the authors do in the adiabatic protocols), the QFI should always be smaller than $t^2\|H_2\|^2$, where H_2 is from Eq. (5), see [Phys. Rev. X 8, 021022 (2018), Ref. [21] of the paper] as well as [PRL 98, 090401 (2007), Ref. [9] of the paper]. This is an important point to clarify: despite the exponential growth, all protocols satisfy the (generalized) Heisenberg limit.

Response: We have discussed the connection between this bound and our results in different parts of the paper. In addition, Fig. 9 has been included to the “Methods” section to show that our results obey the fundamental bounds. Indeed, the exponential enhancement that we observe is rooted in one of the components, namely energy gap (Δ), of these bounds. Regarding the bound of $t^2\|H_2\|^2$, one may notice that that time t needed for probe preparation is proportional to $1/\Delta$. Therefore, the exponentially small energy gap Δ becomes the source of exponential scaling of the QFI. Even by incorporating time as a resource, for which the figure of merit becomes F^Q/T (with T being the preparation time), the bound for the rescaled quantity still goes as $\sim T\|H_2\|^2 \sim \|H_2\|^2/\Delta$. The presence of the energy gap in the denominator shows the origin of the exponential advantage. As a result, the bounds are not only fully satisfied by our findings but also scales similarly with them, clarifying the origin of exponential advantage.

In the revised manuscript, we discuss this issue at several relevant places, including: some discussion in the newly added section “Fundamental QFI bounds in many-body probes”, at the end of the sections where we discuss the QFI of the three models as well as in the section “Resource analysis”.

Reviewer Comment: Second, when time is not considered a resource, the fundamental limits of the QFI of ground-state metrology have recently been established in [PRL 134, 010801 (2025)], where the QFI has been bounded by $\|H_2\|^2/\Delta^2$ (see Eq. (18)). The present results should also satisfy this bound, as the exponential growth of the QFI arises due to the exponential decay of the gap. This also agrees well with the intuition of the authors that the QFI diverges because the gap closes.

Response: We thank the reviewer again for pointing us towards this seminal paper in the context of many-body quantum probes. As mentioned above, our results fully satisfy these fundamental bounds. These have been clarified in the paper, for all the three models, and also with Fig. 9 in the “Methods” section. Interestingly, the bound always show similar scaling with our results, though the prefactor is usually different. Nonetheless, as we analytically show in the “Methods” section, the Grover model at the critical point can even saturate the bound for large system sizes.

Reviewer Comment: Summarizing, despite the exponential growth, the QFI is still bounded by physical limits: both the generalized Heisenberg limit (when time is a resource) and by the recent QFI bound of PRL 134, 010801 (2025). The paper would gain in clarity if the protocols are contrasted with such bounds, and it is explained why and in which precise sense the Heisenberg limit is not violated.

Response: We thank the reviewer for this very important comment. Indeed, we benefited a lot by comparing our results with the fundamental bounds which indicate the origin of exponential scaling that we observe. All the references pointed out by the reviewer are included and we have significantly revised the paper, along with adding a new Fig. 9 to address this issue. Based on this, we hope that the reviewer considers the revised manuscript fit for publication.

Manuscript NCOMMS-24-69490

Exponentially-enhanced quantum sensing with many-body phase transitions

by S. Sarkar, A. Bayat, S. Bose, and R. Ghosh

Point-by-point response to the reviewers' comments

We are greatly thankful to the reviewers for recommending our manuscript for publication. We provide point-by-point response to the comments of the reviewers in this document.

The summary of all the changes is the following:

- A sentence has been added in the third paragraph of the ‘Introduction’ as per the suggestion of reviewer 1.
- The ‘Implementation’ section has been modified to include the issues raised by reviewer 2.
- The typos pointed out by reviewer 2 have been rectified in the ‘Resource analysis’ and the ‘Methods’ sections.
- Another typo has been corrected in the caption of Fig. 8.

Responses to the report of reviewer 1

Reviewer Comment: I believe the Authors have satisfactorily replied to my concerns and to those of the other referees. I think the observation of Referee 2 that the exponential enhancement effectively comes from an exponential increase in the time is an important one for the understanding of the result. While the Authors seem to agree with it, it might be worth adding a sentence of this type in the introduction together with the fact that the current result still satisfies the t^2 bound. I think this would be a very simple, yet important, addition to the paper to avoid future miscomprehensions.

Response: We thank the reviewer for the positive assessment on the revised manuscript and for the important suggestion regarding the enhancement and the bound. We have now added a sentence to this effect in the third paragraph of the ‘Introduction’.

Responses to the report of reviewer 2

Reviewer Comment: I carefully read the revised version. The authors did a good job filling-in many holes in the discussion. I am still considering such a protocol as not very realistic, as exponential precision is steaming from exponentially-long evolution times (the authors use this very argument in paragraph 2 to argue against some non-Hermitian systems proposals), where a lot of control parameters appear to require similar degree of control for the proposal to be successful.

I'm however willing to accept the argument and the proposal.

Response: We thank the reviewer for willing to accept the revised manuscript.

Reviewer Comment: I'm still finding D-Wave proposal as overly optimistic. The authors assume long coherent evolution time, control of the driving schedule, possibility to measure at arbitrary point in the quench. None of this is available in the actual machine, while the authors make it sound as if such an experiment could be performed in the available platform (which is not true).

Response: We agree with the reviewer that these issues can hamper the implementation in the actual D-Wave device. For small enough systems, one may get away with the issues of coherence time and scheduling, but the measurement factor still remains (although D-Wave has recently made some developments towards measurements before the completion of annealing). We have revised the ‘Implementation’ section to include the practical issues raised by the referee.

Reviewer Comment: Minor comments: $|\langle GS(t)|\Psi(t)\rangle|^2 < 1 - \epsilon^2$. still seem to have incorrect direction; $1 - \epsilon^2$ should be a lower bound, not an upper bound if one beliefs in Eq. (5) of Ref [57]. Ditto for the save inequality in other places in the text.

Response: We thank the reviewer for pointing this out. We have rectified the related typos in the ‘Resource analysis’ and the ‘Methods’ sections in the revised manuscript.

Responses to the report of reviewer 3

Reviewer Comment: I am satisfied with the response to my questions and the changes in the manuscript, which has considerably improved after revisions. I recommend acceptance.

As a minor remark, I would suggest introducing (7) before (5)-(6), so that the bounds can be expressed as a function of H_2 (avoiding introducing two different notations for the same quantity).

Response: We thank the reviewer for accepting the revised version of the manuscript. We have positioned Eq. (7) later because we wish to first discuss the general results with the aid of Eqs. (5)-(6). We later showcase the results—first for the specific scenario of Eq. (7), and then in the ‘Resource analysis’ section for adiabatic evolutions considered in Eqs. (16) and (18). Therefore, we hope that the reviewer kindly allows us to retain Eqs. (5)-(6) to address the general case first.